

# Characterisation of bacteria isolated from the stingless bee, *Heterotrigona itama*, honey, bee bread and propolis

Mohamad Syazwan Ngalimat[1,2,*], Raja Noor Zaliha Raja Abd. Rahman[1,2], Mohd Termizi Yusof[2], Amir Syahir[1,3] and Suriana Sabri[1,2,*]

[1] Enzyme and Microbial Technology Research Center, Faculty of Biotechnology and Biomolecular Sciences, Universiti Putra Malaysia, Serdang, Selangor, Malaysia
[2] Department of Microbiology, Faculty of Biotechnology and Biomolecular Sciences, Universiti Putra Malaysia, Serdang, Selangor, Malaysia
[3] Department of Biochemistry, Faculty of Biotechnology and Biomolecular Sciences, Universiti Putra Malaysia, Serdang, Selangor, Malaysia
* These authors contributed equally to this work.

Corresponding author
Suriana Sabri, suriana@upm.edu.my

## ABSTRACT

Bacteria are present in stingless bee nest products. However, detailed information on their characteristics is scarce. Thus, this study aims to investigate the characteristics of bacterial species isolated from Malaysian stingless bee, *Heterotrigona itama*, nest products. Honey, bee bread and propolis were collected aseptically from four geographical localities of Malaysia. Total plate count (TPC), bacterial identification, phenotypic profile and enzymatic and antibacterial activities were studied. The results indicated that the number of TPC varies from one location to another. A total of 41 different bacterial isolates from the phyla Firmicutes, Proteobacteria and Actinobacteria were identified. *Bacillus* species were the major bacteria found. Therein, *Bacillus cereus* was the most frequently isolated species followed by *Bacillus aryabhattai*, *Bacillus oleronius*, *Bacillus stratosphericus*, *Bacillus altitudinis*, *Bacillus amyloliquefaciens*, *Bacillus nealsonii*, *Bacillus toyonensis*, *Bacillus subtilis*, *Bacillus safensis*, *Bacillus pseudomycoides*, *Enterobacter asburiae*, *Enterobacter cloacae*, *Pantoea dispersa* and *Streptomyces kunmingensis*. Phenotypic profile of 15 bacterial isolates using GEN III MicroPlate™ system revealed most of the isolates as capable to utilise carbohydrates as well as amino acids and carboxylic acids and derivatives. Proteolytic, lipolytic and cellulolytic activities as determined by enzymatic assays were detected in *Bacillus stratosphericus* PD6, *Bacillus amyloliquefaciens* PD9, *Bacillus subtilis* BD3 and *Bacillus safensis* BD9. *Bacillus amyloliquefaciens* PD9 showed broad-spectrum of antimicrobial activity against Gram-positive and Gram-negative bacteria in vitro. The multienzymes and antimicrobial activities exhibited by the bacterial isolates from *H. itama* nest products could provide potential sources of enzymes and antimicrobial compounds for biotechnological applications.

## INTRODUCTION

Stingless bee nest products, such as honey, bee bread and propolis have bright economic potentials (*Jacobs et al., 2006*; *Jones, 2013*). Honey (*Amin et al., 2018*), bee bread (*Kroyer & Hegedus, 2001*) and propolis (*Bankova, 2005*) have been applied for centuries in traditional medicine, as well as in food diets and supplementary nutrition. Honey is a natural sweet substance produced by bees from floral nectar or secretions of living parts of plants or excretions of plant-sucking insects. Bees collect and transform the raw materials into honey by combining them with the secretion from the bee's salivary glands, where the mixtures then left in the combs to ripen and mature (*Krell, 1996*). Bee bread is the bee pollen generated from pollen grains that bees collect and mix with the secretion from bee's salivary glands or nectar before being preserved and fermented in the storage pot (*Vásquez & Olofsson, 2009*; *DeGrandi-Hoffman, Eckholm & Huang, 2013*). Propolis is a sticky dark-coloured material collected by bees from material that is actively secreted or exuded by plants, which is then mixed with the wax and used in the nest construction (*Ghisalberti, 1979*). In fact, nest products generated from plant-based materials are rich in macromolecules and possess plenty of antimicrobial activities (*Amin et al., 2018*). Stingless bee nest products, such as honey, are rich in antimicrobial activities due to its high sugar concentration, acidity, hydrogen peroxide ($H_2O_2$) and phytochemical compounds, which are particularly unfavourable for growth of several microorganisms (*Sinacori et al., 2014*). Regardless of these antimicrobial activities, some microbes, such as bacteria, can still be found in honey (*Yaacob et al., 2018*), pollen (*Gilliam, Roubik & Lorenz, 1990*) and bee bread (*Combey, 2017*) of stingless bees.

Bacteria associated with bees have been hypothesised to be involved in the formation of nest products and inhibition of spoilage microorganisms in the storage pots (*Roubik, 1989*; *Gilliam, Roubik & Lorenz, 1990*; *Gilliam, 1997*; *Mueller et al., 2005*; *Anderson et al., 2011*). In fact, nest products can be a good source for the isolation of beneficial bacteria. There are many bacterial genera associated with the nest products such as *Bacillus* and *Streptomyces*, which are characterised as potential biocontrol agent against pathogenic microorganisms. *Bacillus* species isolated from the honey of honeybee, *Apis melifera*, act as a biocontrol agent against the causing agent of chalkbrood disease, *Ascosphaera apis* (*Reynaldi, De Giusti & Alippi, 2004*), and peach gummosis disease, *Botrysphaeria dothidae* (*Li et al., 2016*). The inhibitory activities of *Streptomyces* isolated from *Trigona laeviceps* and *T. fuscobalteata* nests were reported to inhibit the causing agent of American foulbrood disease, *Paenibacillus larvae*, and European foulbrood disease, *Melisococcus plutonius* (*Kroiss et al., 2010*). Moreover, bacterial species isolated from the honey of honeybee were found to produce antimicrobial compounds (*Zhao et al., 2013*; *Obakpororo, Swaroopa & Prakash, 2017*) and enzymes including lipase, protease (*Disayathanoowat et al., 2012*) and amylase (*Wang et al., 2015*).

Investigations of bacteria associated to stingless bee nest products have been focused mostly on the lactic acid bacteria (LAB) isolation and identification for novel LABs with probiotic properties (*Vasquez et al., 2012*; *Leonhardt & Kaltenpoth, 2014*; *Tamarit et al., 2015*; *Yaacob et al., 2018*). However, there are also other types of bacteria besides LABs

with beneficial properties that need to be investigated. Bacterial genus, such as *Bacillus*, are found as the major bacterial genus that can be isolated from varieties of stingless bee species (*Leonhardt & Kaltenpoth, 2014*). Some *Bacillus* species associated with stingless bees were isolated from *Melipona panamica* (*Bacillus alvei* and *Bacillus circulans*) and *T. necrophaga* (*Bacillus circulans*, *Bacillus licheniformis*, *Bacillus megaterium*, *Bacillus pumilus* and *Bacillus subtilis*) nests in Panama (*Gilliam, 1997*). Moreover, other bacterial genera including *Streptomyces* (*Promnuan, Kudo & Chantawannakul, 2009*), *Clostridium*, *Staphylococcus*, *Enterobacter* (*Pucciarelli et al., 2014*), *Ralstonia*, *Pantoea*, *Neisseria*, *Pseudomonas* (*Leonhardt & Kaltenpoth, 2014*), *Lysinibacillus* (*Shanks et al., 2017*) and *Fructobacillus* (*Yaacob et al., 2018*) also have be found associated with stingless bees.

In this work, our aim was to isolate and characterise the bacteria (other than LAB) from *Heterotrigona itama* honey, bee bread and propolis. To date, little is known about the bacterial species (other than LAB) in nest products of stingless bee species, especially for the Malaysian stingless bee, *H. itama*. The other bacterial type might contribute beneficial properties of nest products and need to be explored. Hence, concerted efforts into the identification and characterisation of bacterial species in stingless bee nest products are necessary and merit scientific attention. Bacterial phenotypic profile and proteolytic, lipolytic and cellulolytic activities were determined. In addition, the antimicrobial activities of the bacterial isolates against Gram-positive and Gram-negative bacteria in vitro were investigated. The study on the characteristics of bacterial species from stingless bee nest products is the starting point to elucidate their contributions in the stingless bee nest products.

## MATERIALS AND METHODS

### Collection of samples and bacterial isolation

Propolis, honey and bee bread of *H. itama* nest products were collected from established stingless bee farms in different areas of Malaysia: Yayasan Al-Jenderami (2°52′N, 101°43′E), Ladang nangka PASFA (3°72′N, 103°06′E), Giant B Farm (2°22′N, 102°31′E) and Ladang 10, UPM (2°99′N, 101°71′E). Propolis and bee bread were collected using sterile scissor and spatula, while honey was collected using sterile syringe. All samples were then transferred into sterile 50-ml Falcon tube and stored on ice during transportation to the laboratory for bacterial analysis.

Bacterial isolation was conducted according to the method described by *Yaacob et al. (2018)* with slight modifications. One gram of propolis, honey and bee bread were suspended separately onto 10 ml of nutrient broth (Merck, Darmstadt, Germany), and the mixture was homogenised by vortexing for 10 s. The mixture was then incubated at 37 °C with shaking at $150 \times g$ for 24 h. A 100-μl aliquot of each sample was spread onto nutrient agar medium (Merck, Darmstadt, Germany). The plates were incubated aerobically at 37 °C for approximately 1–3 days. The direct count method for total plate count (TPC) was used to enumerate the amount of viable bacteria (aerobic mesophilic bacteria) in the nest products (*Cappuccino & Sherman, 2011*). Pure bacterial isolates were grown on nutrient agar and incubated at 37 °C for 24 h. Bacterial isolates were then preserved in nutrient broth using 20% (v/v) glycerol at −80 °C.

## Bacterial identification

### DNA extraction and PCR amplification of 16S rRNA gene

The bacterial isolates were identified by sequence analysis of the 16S rRNA gene. Genomic DNA was isolated from overnight cultures using GF-1 bacterial DNA extraction kit (Vivantis Technologies, Subang Jaya, Malaysia) according to the manufacturer's instructions. The 16S rRNA was amplified by polymerase chain reaction (PCR) using the universal primers 8F (5′-AGAGTTTGATCCTGGCTCAG-3′) and 1492R (5′-ACGGC TACCTTGTTACGACTT-3′), which were used to amplify approximately 1.5-kbp segment of the 16S rRNA gene (*Magray et al., 2011*). Each 50 µl of PCR contained 25 µl of 2× PCR Taq Master Mix (Applied Biological Materials Inc., Richmond, BC, Canada), 0.5 µM of each primer and 100 ng of genomic DNA as a template. Thermal cycling was performed in a G-Storm GS1 thermal cycler (GRI Ltd., Essex, UK) with the following parameters: initial denaturation step of 94 °C for 3 min, followed by 35 cycles of 94 °C for 30 s, 58 °C for 30 s and 72 °C for 2 min. A final extension step consisting of 72 °C for 5 min was included. Amplification products were checked by 1.5% (w/v) agarose gel electrophoresis. PCR products were then sequenced (MyTACG Bioscience Enterprise, Kuala Lumpur, Malaysia). The sequences were checked and edited with Chromas Lite software (version 2.6.4; Technelysium Pty Ltd., South Brisbane, QLD, Australia) and compared against the sequences in the National Centre for Biotechnology Information (NCBI) nonredundant database by using the BLASTn program (https://www.ncbi.nlm.nih.gov/).

### Phylogenetic analysis

Multiple alignments of nucleotide gene sequences were created using the program ClustalX (*Thompson et al., 1997*) and MEGA 7.0 software (*Kumar, Stecher & Tamura, 2016*). The neighbour-joining method (*Saitou & Nei, 1987*) with p-distance method (*Nei & Kumar, 2000*) was used to construct phylogenetic trees. The robustness of individual branches was estimated by bootstrapping with 1,000 replications (*Felsenstein, 1985*).

## Biolog GEN III MicroPlate™ system

Phenotypic profile of the bacterial isolates was analysed using GEN III MicroPlates™ (Biolog, Hayward, CA, USA), which includes 94 phenotypic tests, 71 carbon source utilisation assays and 23 chemical sensitivity assays according to the manufacturer's protocol. Bacterial isolates were suspended in the inoculating fluid B (Biolog, Hayward, CA, USA) standardised to 85% T using turbidimeter (Biolog, Hayward, CA, USA). The cell suspension (100 µl) was inoculated into each well of the GENIII MicroPlates™. The plates were then incubated at 30 °C for 24 h. The utilisation pattern was indicated by the reduction of the tetrazolium salt, which is a redox indicator dye that changes from colourless into purple in the well. The colour changes were monitored as absorbance with OmniLog® Incubator/Reader (Biolog, Hayward, CA, USA) at 590 nm. The data were collected using OmniLog® MiroArray™ Data Collection Software 1.2 (Biolog, Hayward, CA, USA).

## Determination of extracellular enzyme activities

### Proteolytic assay

Proteolytic activity was screened using skimmed milk agar and further quantitated using methods described by *Rahman et al. (1994)* with slight modifications. Bacterial isolates were cultured at 37 °C with $150 \times g$ shaking in the production medium (pH 7.0) that composed of trypticase soy broth (Oxoid Ltd., Basingstoke, UK) supplemented with 1% (w/v) tryptone (Oxoid, England). Overnight broth culture was centrifuged at $8,000 \times g$ for 15 min at 4 °C. Supernatant (0.1 ml) was then mixed with one ml of 0.5% (w/v) azocasein (R&M Chemicals, Essex, UK) in 0.1M Tris–HCl (pH 7.0) at 37 °C for 30 min. The reaction was terminated by adding 1.1 ml of 10% (w/v) trichloroacetic acid (Sigma, Ronkonkoma, NY, USA) and incubated at room temperature for 30 min followed by centrifugation at $13,000 \times g$ for 10 min. The resulting supernatant (0.7 ml) was mixed with 0.7 ml of 1M NaOH (R&M Chemicals, UK). Absorbance was then measured using a microplate reader (BioTek Instruments, Winooski, VT, USA) based on the hydrolysis of azocasein by proteases, resulting in release of azo-molecule with a unique absorption at 450 nm. One unit (U) of proteolytic activity is defined in the assay conditions, giving an increase of 0.001 absorbance unit at 450 nm per min. The enzyme assays were performed in triplicates.

### Lipolytic assay

Lipolytic activity was assessed using tributyrin agar and further quantitated using the method described in *Kanwar et al. (2005)* with slight modifications. Bacterial isolates were cultured at 37 °C with $150 \times g$ shaking in the production medium (pH 7.0) that composed of trypticase soy broth, 1% (v/v) olive oil (Bertolli, Lucca, Tuscany, Italy), 1% (w/v) yeast extract (BD, Franklin Lakes, NJ, USA) and 0.5% (w/v) $CaCl_2$ (Merck, Darmstadt, Germany). Overnight broth culture was centrifuged at $8,000 \times g$ for 15 min at 4 °C. Supernatant (0.02 ml) was then mixed with 0.88 ml of 0.1M phosphate buffer (pH 7.0) followed by 0.8 ml of freshly prepared 0.02M p-nitrophenyl palmitate (R&M Chemicals, UK) in isopropanol (R&M Chemicals, UK) at 37 °C and $150 \times g$ shaking for 10 min. The reaction was terminated by adding 0.1 ml of 90% (v/v) ethanol (Systerm, Shah Alam, Malaysia). Absorbance was then measured at 410 nm. The calibration curve was prepared using p-nitrophenol as standard (*Margesin, 2005*). One unit of lipase (U) was defined as the amount of enzyme that releases one μmol of p-nitrophenol per minute under the specified assay conditions described above. The enzyme assays were performed in triplicates.

### Cellulolytic assay

Cellulolytic activity was assessed using carboxymethyl cellulose (CMC) agar and further quantitated using the dinitrosalicylic acid (DNS) method described in *Liang et al. (2014)* with slight modifications. Bacterial isolates were cultured at 37 °C with $150 \times g$ shaking in the production medium (pH 7.0) that composed of trypticase soy broth, 2% (w/v) CMC (R&M Chemicals, UK) and 1% (w/v) yeast extract. Overnight broth culture was centrifuged at $8,000 \times g$ for 15 min at 4 °C. Supernatant (0.05 ml) was then mixed with 0.45 ml of 1% (w/v) CMC in 0.1M phosphate buffer (pH 7.0) at 37 °C and $150 \times g$ shaking for 30 min. The reaction was terminated by adding 0.5 ml of DNS reagent

(1% (w/v) 3,5-dinitrosallicylic acid (R&M Chemicals, Essex, UK), 20% (v/v) 2M NaOH and 30% (w/v) sodium potassium tartrate (R&M Chemicals, Essex, UK)) followed by incubation at 100 °C for 15 min. Absorbance was then measured at 540 nm. The calibration curve was prepared using glucose as standard (*Ghose, 1987*). One unit (U) of the cellulolytic activity was defined as the amount of enzyme that releases one μmol of reducing sugars (measured as glucose) per ml per minute. The enzyme assays were performed in triplicates.

## Determination of antimicrobial activity

Agar-well diffusion assay was carried out according to the method described in *Yilmaz, Soran & Beyatli (2006)* with slight modifications. Bacterial isolates were cultured onto 25 ml Mueller-Hinton broth medium (Merck, Darmstadt, Germany) and incubated at 37 °C for 16 h. Mueller-Hilton agar (Merck, Darmstadt, Germany) plates were swabbed with 100 μl of test bacterial suspension (*Bacillus cereus*, *Staphylococcus aureus*, *Micrococcus luteus*, *Escherichia coli*, *Enterobacter aerogenes*, *Alcaligenes faecalis*, *Aeromonas hydrophila* and *Salmonella typhimurium*) standardised to 0.5 McFarland (R&M Chemicals, UK). The wells of six mm diameter were cut on the agar using the back of a sterile one-ml tip and filled with 100 μl of supernatant of each isolates obtained by centrifugation at $6,000 \times g$ for 15 min at 4 °C. The supernatant was left to dry, and the plate was incubated at 37 °C for 24 h. The diameter of inhibition zone was measured with calipers. In a separate trial, the inhibition zone of chloramphenicol (30 μg; Sigma, Ronkonkoma, NY, USA), which is a broad range antibacterial agent, was determined. The antimicrobial activity assays were performed in triplicates.

# RESULTS

## Bacterial isolation

The TPC was determined after 1–3 days of plate incubation to promote the growth of cultivable bacteria. The TPC range varied from one location to another (Table S1). Therein, the nest products originated from Yayasan Al-Jenderami showed the highest TPC values followed by Giant B Farm, Ladang nangka PASFA and Ladang 10 UPM. The TPC of propolis ranged from $6.3 \times 10^3 \pm 5.5 \times 10^2$ to $1.8 \times 10^4 \pm 5.8 \times 10^4$ colony forming units per gram (cfu/g), while those recorded for honey samples ranged from 0.0 to $8.0 \times 10^3 \pm 1.0 \times 10^3$ cfu/g. The TPC of bee bread samples ranged from 0.0 to $8.6 \times 10^3 \pm 5.8 \times 10^3$ cfu/g. Among the three nest products, propolis showed the highest TPC value ($1.8 \times 10^4 \pm 5.8 \times 10^4$ cfu/g) while the lowest were honey (0.0 cfu/g) and bee bread (0.0 cfu/g). According to the results, 41 isolates were recovered from *H. itama* nest products. As analysed by Gram and endospore staining, as well as catalase test, 37 of the isolates were preliminarily classified as bacteria from phylum Firmicutes, while there were three from phylum Proteobacteria and one from phylum Actinobacteria (Table S2).

## Identification of bacterial isolates

To identify the bacterial isolates, approximately 1.5-kbp fragment of the 16S rRNA gene was amplified from the isolates' genomic DNA. The amplified fragments were compared to those sequences deposited in the GenBank database (Table S3). The 16S rRNA gene

sequence of the bacterial isolates displayed a similarity of ≥99% to the closest known species. Based on the 16S rRNA gene, most of the isolates belonged to the phylum Firmicutes (*Bacillus* spp.) followed by Proteobacteria (*Enterobacter* spp. and *Pantoea* sp.) and Actinobacteria (*Streptomyces* sp.) (Fig. 1). The 37 bacterial isolates belonging to the phylum Firmicutes were *Bacillus cereus* (19), *Bacillus aryabhattai* (5), *Bacillus oleronius* (2), *Bacillus stratosphericus* (2), *Bacillus altitudinis* (2), *Bacillus amyloliquefaciens* (2), *Bacillus nealsonii* (1), *Bacillus toyonensis* (1), *Bacillus subtilis* (1), *Bacillus safensis* (1) and *Bacillus pseudomycoides* (1). The three bacterial isolates belonging to the phylum Proteobacteria were *Enterobacter asburiae* (1), *Enterobacter cloacae* (1) and *Pantoea dispersa* (1), while the one bacterial isolate from the phylum Actinobacteria was *Streptomyces kunmingensis*. From the list, 15 bacterial isolates of different identified species named *Bacillus cereus* HD1, *Bacillus aryabhattai* BD8, *Bacillus oleronius* PD3, *Bacillus stratosphericus* PD6, *Bacillus altitudinis* BD4, *Bacillus amyloliquefaciens* PD9, *Bacillus nealsonii* PD4, *Bacillus toyonensis* PD13, *Bacillus subtilis* BD3, *Bacillus safensis* BD9, *Bacillus pseudomycoides* HM2, *Enterobacter asburiae* PD12, *Enterobacter cloacae* PM4, *Pantoea dispersa* PG1 and *Streptomyces kunmingensis* BG1 were selected for further analysis on their phenotypic profile and enzymatic and antimicrobial activities.

## Phenotypic profile of bacterial isolates

GEN III MicroPlates™ assay from Biolog is commonly used for bacterial identification. However, in this study, it was used to determine the phenotypic profile of the isolates based on the utilisation of carbohydrates, amino acids and carboxylic acids that are available on the plate. According to the results, each isolate has a different utilisation pattern for carbohydrates as well as amino acids, carboxylic acids and their derivatives (Table 1). Moreover, most of the isolates were able to utilise monosaccharides (α-D-glucose, D-fructose, D-mannose, D-galactose and D-fucose), disaccharides (D-cellobiose, gentiobiose, sucrose, trehalose, D-maltose, D-turanose, α-D-lactose and D-melibiose) and polymers (dextrin and pectin). Notably, *Enterobacter asburiae* PD12, *Enterobacter cloacae* PM4 and *Pantoea dispersa* PG1 showed utilisation of *N*-acetyl-D-glucosamine and *N*-acetyl-β-D-mannosamine.

*Bacillus cereus* HD1 and *Enterobacter asburiae* PD12 showed the highest utilisation of tested amino acids and derivatives (72.7%). There was no utilisation of amino acids and derivatives shown by *Bacillus oleronius* PD3, *Bacillus nealsonii* PD4 and *Streptomyces kunmingensis* BG1. In addition, most of the isolates were able to use non-essential amino acids, including alanine, aspartic acid and glutamic acid, as their nitrogen source. Of all isolates, only *Bacillus oleronius* PD3, *Bacillus nealsonii* PD4 and *Streptomyces kunmingensis* BG1 were able to utilise L-alanine and L-glutamic acid. On the other hand, *Bacillus toyonensis* PD13 showed the highest utilisation of carboxylic acids and derivatives tested (61.1%), while the lowest utilisation was detected from *Bacillus oleronius* PD3 (5.6%). Furthermore, most of the isolates were able to use citric acid, L-malic acid, acetoacetic acid, L-lactic acid and acetic acid. No utilisation of D-lactic acid methyl ester and α-keto-butyric acid were detected from any of the isolates.

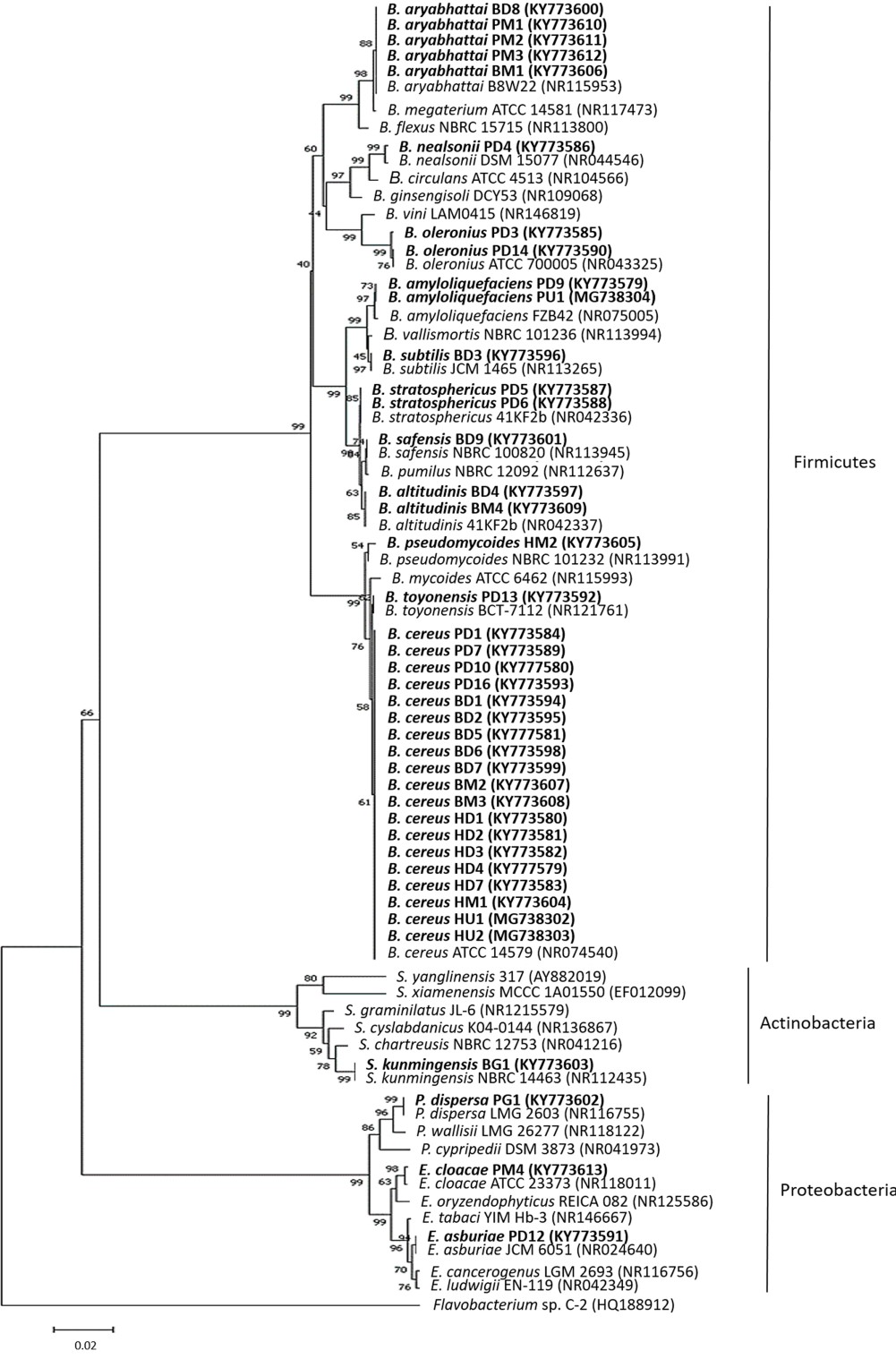

**Figure 1 The 16S rRNA gene phylogenetic tree analysis of the bacterial isolates.** The evolutionary history was inferred using the neighbour-joining method. The isolates were aligned with closely related strains from GenBank. The accession numbers indicated within bracket. The optimal tree with the sum of branch length = 0.71705651 was shown. The analysis involved 76 nucleotide sequences. There were a total of 1,087 positions in the final dataset. Sequence from *Flavobacterium* sp. C-2 was used as an out group. The bolded text indicates bacteria isolated in this study. *B*, *Bacillus*; *E*, *Enterobacter*; *P*, *Pantoea* and *S*, *Streptomyces*.             

**Table 1  Phenotypic profile of bacterial isolates.**

| Substrates | B. cereus HD1 | B. aryabhattai BD8 | B. oleronius PD3 | B. stratosphericus PD6 | B. altitudinis BD4 | B. amyloliquefaciens PD9 | B. nealsonii PD4 | B. toyonensis PD13 | B. subtilis BD3 | B. safensis BD9 | B. pseudomycoides HM2 | E. asburiae PD12 | E. cloacae PM4 | P. dispersa PG1 | S. kunmingensis BG1 |
|---|---|---|---|---|---|---|---|---|---|---|---|---|---|---|---|
| **Percentage (%) well showed utilisation of carbohydrates (n = 42)** | | | | | | | | | | | | | | | |
| | 40.5 | 19.1 | 23.8 | 35.7 | 33.3 | 52.4 | 7.1 | 38.1 | 54.8 | 59.5 | 33.3 | 78.6 | 81.0 | 71.4 | 9.5 |
| Dextrin | + | + | + | + | + | + | + | + | + | + | + | + | + | + | + |
| D-trehalose | + | + | − | + | + | + | − | + | + | + | + | + | + | + | + |
| β-methyl-D-glucoside | + | − | − | + | + | + | − | + | + | + | + | + | + | + | + |
| α-D-glucose | + | − | − | + | + | + | − | + | + | + | + | + | + | + | + |
| D-cellobiose | + | − | − | + | − | + | − | + | + | + | + | + | + | + | + |
| Gentiobiose | − | − | + | + | + | + | − | + | + | + | − | + | + | + | − |
| Sucrose | + | + | − | + | + | + | − | + | + | + | − | + | + | + | + |
| D-fructose | + | − | + | + | − | + | − | + | + | + | − | + | + | + | + |
| D-maltose | + | + | − | − | − | + | − | + | + | + | + | + | − | − | + |
| D-mannose | + | − | + | − | + | + | − | − | + | + | − | + | + | + | + |
| N-acetyl-D-glucosamine | + | + | + | − | + | + | − | + | − | − | + | + | + | + | + |
| D-salicin | − | + | + | + | − | + | − | − | + | + | − | + | + | − | − |
| D-raffinose | − | − | − | − | − | + | − | − | + | + | − | + | + | − | − |
| D-galactose | − | − | − | − | − | + | − | − | − | − | + | + | + | + | − |
| Inosine | + | − | − | − | − | + | − | + | − | − | − | + | + | + | + |
| D-turanose | − | + | + | − | − | + | − | − | + | + | − | − | − | − | − |
| Stachyose | − | + | + | − | − | + | − | − | − | − | − | + | − | − | − |
| D-melibiose | + | − | − | − | − | − | − | − | − | − | − | + | + | + | + |
| N-acetyl-β-D-mannosamine | − | − | − | − | − | − | − | − | − | − | − | + | + | + | − |
| α-D-lactose | − | − | − | − | − | + | − | − | − | − | − | + | − | − | − |
| L-rhamnose | − | − | − | − | − | − | − | − | − | − | − | − | + | + | − |
| N-acetyl-D-galactosamine | − | − | − | − | − | − | − | − | − | − | − | + | + | − | − |
| N-acetyl neuraminic acid | − | − | − | − | − | − | − | − | − | − | − | + | − | − | − |
| D-fucose | − | − | − | − | − | − | − | − | − | − | − | − | − | + | − |
| 3-methyl glucose | − | − | − | − | − | − | − | − | − | − | − | − | − | + | − |
| L-fucose | − | − | − | − | − | − | − | − | − | − | − | − | − | − | − |
| Glycerol | + | − | + | + | + | + | − | + | + | + | + | + | + | + | + |
| D-mannitol | − | − | + | + | + | + | − | − | + | + | + | + | + | + | − |
| D-sorbitol | − | − | − | − | − | + | − | − | + | + | + | + | + | − | − |
| Myo-inositol | − | − | − | − | − | − | − | − | − | − | − | + | + | + | − |
| D-arabitol | − | − | − | − | − | − | + | − | − | − | − | − | − | + | − |
| D-fructose-6-phosphate | + | + | + | + | + | − | − | + | + | + | + | + | + | + | + |
| D-glucose-6-phosphate | + | − | − | − | − | − | − | − | − | − | + | + | + | + | + |

(Continued)

| Substrates | B. cereus HD1 | B. aryabhattai BD8 | B. oleronius PD3 | B. stratosphericus PD6 | B. altitudinis BD4 | B. amyloliquefaciens PD9 | B. nealsonii PD4 | B. toyonensis PD13 | B. subtilis BD3 | B. safensis BD9 | B. pseudomycoides HM2 | E. asburiae PD12 | E. cloacae PM4 | P. dispersa PG1 | S. kunmingensis BG1 |
|---|---|---|---|---|---|---|---|---|---|---|---|---|---|---|---|
| Pectin | + | − | + | + | + | − | − | + | + | + | + | + | + | + | + |
| Glucuronamide | − | − | + | + | + | + | + | − | − | + | + | + | + | + | − |
| D-gluconic acid | + | + | + | − | − | − | − | + | + | + | − | + | + | + | + |
| D-galacturonic acid | − | − | − | − | − | − | − | + | + | + | − | + | + | + | − |
| L-galactonic acid lactone | − | − | − | − | − | − | − | − | + | + | − | + | + | + | − |
| D-glucuronic acid | − | − | − | − | − | − | − | + | − | + | − | + | + | + | − |
| D-saccharic acid | − | − | − | − | − | − | − | − | + | + | − | + | + | + | − |
| Mucic acid | − | − | − | − | − | − | − | − | + | + | − | + | + | + | − |
| Quinic acid | − | − | − | + | + | − | − | − | − | − | − | − | − | + | − |
| **Percentage (%) well showed utilisation of amino acids and derivatives (n = 11)** | 72.7 | 45.5 | 0.0 | 45.5 | 36.4 | 54.5 | 0.0 | 45.5 | 27.3 | 27.3 | 45.5 | 72.7 | 63.6 | 54.5 | 0.0 |
| L-alanine | + | + | − | + | + | + | − | + | + | + | + | + | + | + | − |
| L-glutamic acid | + | + | − | + | + | + | − | + | + | + | + | + | + | + | − |
| L-aspartic acid | + | + | − | + | + | + | − | + | + | + | − | + | + | + | − |
| L-histidine | + | + | − | − | − | + | − | + | − | − | + | + | + | + | − |
| L-serine | + | − | − | + | − | − | − | − | − | − | + | + | + | + | − |
| Glycyl-L-proline | + | − | − | − | − | + | − | − | − | − | − | + | + | + | − |
| Gelatin | + | − | − | − | − | + | − | + | − | − | + | − | − | − | − |
| D-serine | + | + | − | − | − | − | − | − | − | − | − | + | + | − | − |
| D-aspartic acid | − | − | + | + | + | − | − | − | − | − | − | − | − | − | − |
| L-arginine | − | − | − | − | − | − | − | − | − | − | − | + | − | − | − |
| L-pyroglutamic acid | − | + | − | − | − | − | − | − | − | − | − | − | − | − | − |
| **Percentage (%) well showed utilisation of carboxylic acids and derivatives (n = 18)** | 55.6 | 55.6 | 5.6 | 16.7 | 11.1 | 22.2 | 11.1 | 61.1 | 16.7 | 27.8 | 33.3 | 44.4 | 50.0 | 44.4 | 22.2 |
| Citric acid | + | + | − | + | + | + | − | + | + | + | + | + | + | + | − |
| L-malic acid | + | + | − | + | + | + | − | + | + | + | + | + | + | + | − |
| Acetoacetic acid | + | + | + | − | − | − | + | + | − | − | − | + | + | + | + |
| L-lactic acid | + | + | − | − | − | + | − | + | + | + | − | + | + | + | − |
| Acetic acid | + | + | − | − | − | − | + | + | − | − | − | + | + | + | + |
| Formic acid | + | + | − | − | − | − | − | + | − | + | + | − | − | + | − |
| Methyl pyruvate | + | − | − | − | − | − | − | + | − | − | + | + | + | − | − |
| Bromo-succinic acid | + | + | − | + | − | − | − | − | − | − | − | + | + | + | − |
| Tween 40 | + | + | − | − | − | − | − | + | − | − | + | − | − | − | + |
| β-hydroxy-D,L-butyric acid | + | + | − | − | − | − | − | + | − | − | − | − | + | − | − |
| γ-amino-butyric acid | − | − | − | + | − | − | − | + | − | − | − | − | − | + | − |

| Substrates | B. cereus HD1 | B. aryabhattai BD8 | B. oleronius PD3 | B. stratosphericus PD6 | B. altitudinis BD4 | B. amyloliquefaciens PD9 | B. nealsonii PD4 | B. toyonensis PD13 | B. subtilis BD3 | B. safensis BD9 | B. pseudomycoides HM2 | E. asburiae PD12 | E. cloacae PM4 | P. dispersa PG1 | S. kunmingensis BG1 |
|---|---|---|---|---|---|---|---|---|---|---|---|---|---|---|---|
| Propionic acid | – | + | – | – | – | – | – | – | – | – | – | – | – | – | + |
| p-hydroxy phenylacetic acid | – | – | – | – | – | – | – | – | – | – | – | + | + | – | – |
| α-keto-glutaric acid | – | – | – | – | – | – | – | – | – | – | + | – | – | – | – |
| D-malic acid | – | – | – | – | – | + | – | – | – | – | – | – | – | – | – |
| α-hydroxy butyric acid | – | – | – | – | – | – | – | + | – | – | – | – | – | – | – |
| D-lactic acid methyl ester | – | – | – | – | – | – | – | – | – | – | – | – | – | – | – |
| α-keto-butyric acid | – | – | – | – | – | – | – | – | – | – | – | – | – | – | – |
| **Percentage (%) well showed chemicals resistancy (n = 23)** | 56.5 | 39.1 | 47.8 | 43.5 | 43.5 | 52.2 | 43.5 | 43.5 | 52.2 | 52.2 | 47.8 | 78.3 | 62.2 | 52.2 | 17.4 |
| pH 7 | + | + | + | + | + | + | + | + | + | + | + | + | + | + | + |
| pH 6 | – | + | + | + | + | + | + | – | + | + | – | + | + | + | + |
| pH 5 | + | – | – | – | – | + | – | + | + | + | + | + | – | – | + |
| 1% NaCl | + | + | + | + | + | + | + | + | + | + | + | + | + | + | – |
| 4% NaCl | + | + | + | + | + | + | + | + | + | + | + | + | + | – | – |
| 8% NaCl | + | + | + | + | + | + | – | – | + | + | + | + | – | – | – |
| 1% sodium lactate | + | + | + | + | + | + | + | + | + | + | – | + | + | + | – |
| Lithium chloride | + | + | + | + | + | + | + | + | + | + | + | + | + | – | – |
| Potassium tellurite | + | + | + | + | + | + | + | + | + | + | + | – | – | – | + |
| Guanidine HCl | + | – | + | + | + | + | + | + | – | – | + | + | + | + | – |
| Aztreonam | + | + | + | + | + | + | + | + | + | + | + | – | – | – | – |
| Sodium butyrate | + | – | + | – | – | + | – | + | + | + | + | + | + | + | – |
| Rifamycin SV | – | – | – | – | – | – | + | – | – | + | – | + | + | + | – |
| Sodium bromate | + | + | + | + | + | + | – | – | + | – | + | – | – | – | + |
| Tetrazolium violet | – | – | – | – | – | – | – | – | – | – | – | + | + | + | – |
| Tetrazolium blue | – | – | – | – | – | – | – | – | – | – | – | + | + | + | – |
| Troleandomycin | – | – | – | – | – | – | – | – | – | – | – | + | + | + | – |
| Lincomycin | – | – | – | – | – | – | – | – | – | – | – | + | + | + | – |
| Niaproof 4 | – | – | – | – | – | – | – | – | – | – | – | + | + | + | – |
| Vancomycin | – | – | – | – | – | – | – | – | – | – | – | + | + | + | – |
| Fusidic acid | – | – | – | – | – | – | – | – | – | – | – | + | + | – | – |
| Minocycline | + | – | – | – | – | – | – | – | – | – | – | – | – | – | – |
| Nalidixic acid | – | – | – | – | – | – | – | – | – | – | – | – | – | – | – |

**Notes:**
The phenotypic profile of bacterial isolates as determined by Biolog GENIII MicroPlate assay.
+, Positive utilisation and –, negative utilisation. B, *Bacillus*; E, *Enterobacter*; P, *Pantoea* and S, *Streptomyces*.

Among all isolates, bacteria from the phylum Proteobacteria showed the highest resistant to antibiotics and chemicals tested. *Enterobacter asburiae* PD12, *Enterobacter cloacae* PM4 and *Pantoea dispersa* PG1 were resistant to guanidine HCl, rifamycin SV, tetrazolium violet, tetrazolium blue, troleandomycin, lincomycin, niaproof 4 and vancomycin. *Streptomyces kunmingensis* BG1 was found as a highly sensitive isolate. It was detected to be sensitive to lithium chloride, guanidine HCl, aztreonam, sodium butyrate, rifamycin SV, tetrazolium violet, tetrazolium blue, troleandomycin, lincomycin, niaproof 4, vancomycin, fusidic acid, minocycline and nalidixic acid.

## Extracellular enzyme activities of bacterial isolates

Bacterial isolates were screened for cellulase, protease and lipase activities (Table S2). All of the isolates possessed single or multiple enzyme activities (either two or three enzyme activities), except for *Bacillus oleronius* PD3 and *Bacillus nealsonii* PD4 where proteolytic, lipolytic and cellulolytic activities were not detected. The activities were further quantified using enzyme assays (Table 2). Cellulase activity was exhibited by the majority of the isolates (67%) followed by protease (60%) and lipase (33%) (Table S2). Low cellulolytic activities were detected from *Bacillus stratosphericus* PD6, *Bacillus toyonensis* PD13, *Enterobacter cloacae* PM4, *Enterobacter asburiae* PD12, *Streptomyces kunmingensis* BG1 and *Pantoea dispersa* PG1 (ranging from 0.18 to 0.31 U/ml), whereas high cellulolytic activities were found from *Bacillus cereus* HD1, *Bacillus amyloliquefaciens* PD9, *Bacillus safensis* BD9 and *Bacillus subtilis* BD3 (ranging from 0.69 to 0.78 U/ml) (Table 2). The highest proteolytic activity was detected from *Bacillus amyloliquefaciens* PD9 (2.55 U/ml), while the lowest was found from *Bacillus stratosphericus* PD6, *Bacillus subtilis* BD3 and *Bacillus aryabhattai* BD8. Lastly, *Bacillus subtilis* BD3 showed the highest lipolytic activity (0.80 U/ml), and the lowest was found from *Bacillus safensis* BD9 (0.41 U/ml) (Table 2).

## Antimicrobial activity of bacterial isolates

By using cell-free supernatant against tested bacteria to determine the bacterial isolates' antimicrobial activity, we found that only four bacterial isolates, namely *Bacillus cereus* HD1, *Bacillus altitudinis* BD4, *Bacillus amyloliquefaciens* PD9 and *Bacillus safensis* BD9, showed antimicrobial activities. The cell-free supernatant of the *Bacillus amyloliquefaciens* PD9 exhibited broad-spectrum antimicrobial activity towards Gram-positive (*Bacillus cereus*, *Staphylococcus aureus* and *Micrococcus luteus*) and Gram-negative (*Enterobacter aerogenes*, *Escherichia coli*, *Alcaligenes faecalis*, *Aeromonas hydrophila* and *Salmonella typhimurium*) bacteria (Table 3).

## DISCUSSION

Bacteria are proven to be present in the stingless bee nest products. As the interaction of stingless bee and bacteria are still unclear, it is thus important remark to study bacterial characteristics to obtain fundamental knowledge about their metabolic and phenotypic profiles. With the aims to investigate the characteristics of bacterial species present in *H. itama* nest products, the characteristics of bacterial species associated with *H. itama* honey, bee bread and propolis have been explored.
**Table 2 Enzymatic activity of bacterial isolates recovered from *H. itama* nest products.**

| Isolate | Extracellular enzyme activity (U/ml) | | |
|---|---|---|---|
| | Proteolytic | Lipolytic | Cellulolytic |
| *B. cereus* HD1 | 1.93 ± 0.01 | ND | 0.78 ± 0.01 |
| *B. aryabhattai* BD8 | 1.11 ± 0.00 | ND | ND |
| *B. oleronius* PD3 | ND | ND | ND |
| *B. stratosphericus* PD6 | 1.11 ± 0.02 | 0.48 ± 0.02 | 0.31 ± 0.01 |
| *B. altitudinis* BD4 | ND | 0.55 ± 0.02 | ND |
| *B. amyloliquefaciens* PD9 | 2.55 ± 0.02 | 0.78 ± 0.03 | 0.73 ± 0.01 |
| *B. nealsonii* PD4 | ND | ND | ND |
| *B. toyonensis* PD13 | 1.15 ± 0.00 | ND | 0.26 ± 0.01 |
| *B. subtilis* BD3 | 1.11 ± 0.03 | 0.80 ± 0.01 | 0.69 ± 0.00 |
| *B. safensis* BD9 | 1.12 ± 0.01 | 0.41 ± 0.03 | 0.73 ± 0.02 |
| *B. pseudomycoides* HM2 | 1.12 ± 0.01 | ND | ND |
| *E. asburiae* PD12 | ND | ND | 0.21 ± 0.02 |
| *E. cloacae* PM4 | 1.12 ± 0.00 | ND | 0.26 ± 0.01 |
| *P. dispersa* PG1 | ND | ND | 0.18 ± 0.02 |
| *S. kunmingensis* BG1 | ND | ND | 0.19 ± 0.01 |

Notes:
Reported values are mean ± SD of three replicates in duplicate samples.
ND, no detectable activity.

Based on our study, the common group of the bacteria found in *H. itama* nest products were Firmicutes followed by Proteobacteria and Actinobacteria. The majority of the isolates found in the nest products were *Bacillus* species. There were 11 different *Bacillus* spp. which are *Bacillus cereus, Bacillus aryabhattai, Bacillus oleronius, Bacillus stratosphericus, Bacillus altitudinis, Bacillus amyloliquefaciens, Bacillus nealsonii, Bacillus toyonensis, Bacillus subtilis, Bacillus safensis* and *Bacillus pseudomycoides* were isolated and identified. Similar bacterial species namely *Bacillus cereus* strains were isolated from honeybee (*López & Alippi, 2007*), solitary bee (*Gilliam, Roubik & Lorenz, 1990*) and stingless bee (*Pucciarelli et al., 2014*) colonies. Also, *Bacillus aryabhattai* (*Sudhagar, Reddy & Nagalakshmi, 2017*), *Bacillus oleronius* (*Gasper et al., 2017*) and *Bacillus subtilis* (*Sabate, Carrillo & Carina Audisio, 2009*) were found in the gut of honeybees, whereas *Bacillus altitudinis* (*Zhang et al., 2016*), *Bacillus amyloliquefaciens* (*Zhao et al., 2015*) and *Bacillus safensis* (*Sinacori et al., 2014*) were found in the honey. Other than Firmicutes, bacteria from the phylum Proteobacteria and Actinobacteria were also found associated with honeybee (*Mattila et al., 2012; Vojvodic, Rehan & Anderson, 2013; Lee et al., 2015; Tarpy, Mattila & Newton, 2015; Khan et al., 2017*) and stingless bee (*Leonhardt & Kaltenpoth, 2014*). This striking similarity of bacterial content between honeybee and stingless bee may reflect the similar bacterial roles in the bee colonies. Although these bacterial species are known to be associated with bees, their biological functions are poorly understood.

Insects engage in a vast array of symbiotic relationships with a wide diversity of microorganisms; in which some of them benefit the host nutritionally (*Klepzig et al., 2009*). In stingless bee nests, bacteria could be involved in the degradation of nest

**Table 3 Diameters of inhibition zone (mm) exhibited against test bacteria of bacterial isolates and standard antibiotics.**

| Isolate | Microorganisms and inhibition zone (mm) | | | | | | | |
|---|---|---|---|---|---|---|---|---|
| | B.c | S.a | M.l | E.a | E.c | A.f | A.h | S.t |
| B. cereus HD1 | ND | 10.4 ± 0.1 | 13.2 ± 0.0 | ND | ND | ND | ND | ND |
| B. aryabhattai BD8 | ND | ND | ND | ND | ND | ND | ND | ND |
| B. oleronius PD3 | ND | ND | ND | ND | ND | ND | ND | ND |
| B. stratosphericus PD6 | ND | ND | ND | ND | ND | ND | ND | ND |
| B. altitudinis BD4 | ND | ND | 10.7 ± 0.0 | ND | ND | ND | ND | ND |
| B. amyloliquefaciens PD9 | 13.2 ± 0.2 | 13.2 ± 0.0 | 14.3 ± 0.2 | 12.6 ± 0.1 | 15.3 ± 0.2 | 15.3 ± 0.3 | 11.8 ± 0.3 | 11.1 ± 0.1 |
| B. nealsonii PD4 | ND | ND | ND | ND | ND | ND | ND | ND |
| B. toyonensis PD13 | ND | ND | ND | ND | ND | ND | ND | ND |
| B. subtilis BD3 | ND | ND | ND | ND | ND | ND | ND | ND |
| B. safensis BD9 | ND | ND | 10.1 ± 0.1 | ND | ND | ND | ND | ND |
| B. pseudomycoides HM2 | ND | ND | ND | ND | ND | ND | ND | ND |
| E. asburiae PD12 | ND | ND | ND | ND | ND | ND | ND | ND |
| E. cloacae PM4 | ND | ND | ND | ND | ND | ND | ND | ND |
| P. dispersa PG1 | ND | ND | ND | ND | ND | ND | ND | ND |
| S. kunmingensis BG1 | ND | ND | ND | ND | ND | ND | ND | ND |
| Chloramphenicol (30 µg) | 20.1 ± 0.2 | 25.4 ± 0.1 | 32.0 ± 0.0 | 23.5 ± 0.2 | 22.0 ± 0.3 | 20.2 ± 0.1 | 22.7 ± 0.1 | 25.2 ± 0.2 |

**Notes:**
Reported values are mean ± SD of three replicates in duplicate samples.
B.c, B. cereus; S.a, S. aureus; M.l, M. luteus; E.a, E. aerogenes; E.c, E. coli; A.f, A. faecalis; A.h, A. hydrophila; S.t, S. typhimurium; ND, no detectable activity.

products by producing some enzymes (*Gilliam, Roubik & Lorenz, 1990*). Nest products generated from plant-based materials are composed of complex biomolecules, such as cellulose, proteins and lipids. The degradation of those materials requires hydrolytic enzymes. In the enzymatic assays, we have observed that cellulase activity was exhibited by the majority of the isolates, followed by protease and lipase. In the bee nests, the enzymes produced by the bacterial isolates might be involved in the breakdown of complex biomolecules (such as carbohydrates and proteins) that helped in the formation of nest products (*Gilliam, Roubik & Lorenz, 1990*; *Vásquez & Olofsson, 2009*; *Lee et al., 2015*). Thus, it is possible that these bacteria may play roles in food digestion for the stingless bee.

In addition, the main food source of the stingless bee are nectar and pollen grains, which contain various types of carbon source. Evidently, monosaccharides, such as glucose, fructose and mannose, and disaccharides, such as cellobiose, gentiobiose, sucrose and trehalose, are found in various floral nectars (*Wackers, 2001*). Pollen grains contain varieties of amino acids, including alanine, aspartic acid and glutamic acid (*DeGrandi-Hoffman, Eckholm & Huang, 2013*). On the other hand, honey and bee bread are reported to contain varieties of nonaromatic organic acids, including malic, citric, lactic, succinic and fumaric acids (*Sancho et al., 2013*; *Kieliszek et al., 2018*). In this study, the Biolog GEN III MicroPlates™ assay helped us to determine the ability of the isolates to utilise of various sugars, amino acid and carboxylic acids that are normally found in the raw material of honey (nectar) and bee bread (pollen grains).

The antimicrobial activities of bacteria from *H. itama* nest products in Gram-positive and Gram-negative bacteria inhibition also have been investigated. There were four isolates from phylum Firmicutes that showed antimicrobial activities, namely *Bacillus amyloliquefaciens* PD9, *Bacillus altitudinis* BD4, *Bacillus safensis* BD9 and *Bacillus cereus* HD1. Of note, bacteria from the genus *Bacillus* are commonly isolated from honeybee products and found to eliminate the unnecessary microbes that can cause destruction to the bee colony (*Gilliam, Roubik & Lorenz, 1990*). For example, *Bacillus amyloliquefaciens* has been shown to produce antimicrobial compounds, such as lipopeptides, that are active against an important honeybee pathogen, *Paenibacillus larvae* (*Benitez et al., 2012*). These antimicrobial substances inhibit the growth of competing organisms (fungi and other bacteria) (*Cochrane & Vederas, 2014*) and therefore might cause the antibacterial activity against the food spoilage microorganisms in the bee nests.

## CONCLUSIONS

From this study, *H. itama* honey, bee bread and propolis contain bacteria. The limited number of bacteria could be due to the fact that nest products are rich in antimicrobial activities that are attributed to its physicochemical and phytochemical composition. This work forms the foundation for future research to explore the biotechnological potential of bacterial isolates from *H. itama* nest products as extracellular hydrolytic enzymes and antimicrobial compound producers.

## ACKNOWLEDGEMENTS

We thank the Yayasan Al-Jenderami, Ladang nangka PASFA, Giant B Farm and Ladang 10, UPM for providing us with *H. itama* products. We are immensely grateful to Dr. Nurhidayah Roslan and Dr. Zulkifli Mustafa for providing us *H. itama* products and initial guideline for the microbial isolation. Special thanks to Dr. Murni Marlina Abd Karim from Faculty of Agriculture, UPM for providing us *A. hydrophila* culture for the antimicrobial tests.

### Funding

This research is supported by the Universiti Putra Malaysia—Putra Graduate Initiative (GP-IPS/2018/9601400). Mohamad Syazwan Ngalimat is supported by a GRF scholarship from Universiti Putra Malaysia. The funders had no role in study design, data collection and analysis, decision to publish, or preparation of the manuscript.

### Grant Disclosures

The following grant information was disclosed by the authors:
Universiti Putra Malaysia—Putra Graduate Initiative: GP-IPS/2018/9601400.

### Competing Interests

The authors declare that they have no competing interests.

## Author Contributions

- Mohamad Syazwan Ngalimat conceived and designed the experiments, performed the experiments, analysed the data, prepared figures and/or tables, authored or reviewed drafts of the paper, approved the final draft.
- Raja Noor Zaliha Raja Abd. Rahman contributed reagents/materials/analysis tools.
- Mohd Termizi Yusof contributed reagents/materials/analysis tools.
- Amir Syahir contributed reagents/materials/analysis tools.
- Suriana Sabri conceived and designed the experiments, analysed the data, contributed reagents/materials/analysis tools, prepared figures and/or tables, authored or reviewed drafts of the paper, approved the final draft.

## DNA Deposition

The following information was supplied regarding the deposition of DNA sequences:

The isolates 16S rRNA sequences described here are available via GenBank accession numbers KY773579–KY773590, KY773592–KY773601, KY773604–KY773612, KY777579–KY777581 and MG738302–MG738304.

## Data Availability

The raw data is available in Files S1 and S2.

## Supplemental Information

Supplemental information for this article can be found online at http://dx.doi.org/10.7717/peerj.7478#supplemental-information.

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
