# Peer review of "Characterisation of bacteria isolated from the stingless bee, Heterotrigona itama, honey, bee bread and propolis"

_PeerJ, doi:10.7717/peerj.7478_

## Round 0.1 · original submission · Major Revisions

The reviewers had several significant concerns including the lack of a robust rationale or hypothesis for the study (which could be better framed in the introduction and discussion sections).

In addition, you must have the text copy-edited by a professional company or a native English speaker as much of your meaning was obscured by the lack of English proficiency in the writing. This is critical to the acceptance (or otherwise) of your submission.

Reviewer #1 had some especially helpful and detailed critiques, which I suggest you heed.

Reviewer 1 ·

Basic reporting

• The authors do a great job providing background knowledge about the bees, their hive components, the physical properties of the components, and the microbes associated with bees, but never really state a strong hypothesis for conducting the study, besides being interested in characterizing the strains of Bacillus. Do the authors have a hypothesis bout the bacteria providing some type of service to the hive (protection, metabolic breakdown)? Currently it is unclear what the expectations are for the study. Please add text that explicitly states a hypothesis.

• Is there evidence these bacteria are active while present in hive components? Could the bacteria just be dormant (spores) while in the hive components due to a harsh environment, but easily recovered when plated on rich media? Is there any literature on the activity of these bacteria in vivo (this topic seems pertinent to the study, but is never addressed)?

• What is the difference between “pure culture” (line 113) and “pure bacterial isolate” (line 118)? Is there a difference (there seems to be difference based on context)? Please clarify.

• Why is table 1 a main figure in the manuscript? It should be in the supplemental. If anything some of the data from ST3 should be made into a main figure to represent the metabolic data, which seems to be a big part of the paper.

• It would be helpful if the authors order the results and isolates based on their phylogenetic relatedness for all relevant tables (tables 2, 3, ST3, Files 1 and 2)

• Are file 1 and 2 supplemental tables? Or are they just files that will be deposited with the manuscript?

• For TS3, it would be nice to know what range (raw values) in which the authors are determining to be “+, -, and +/-“. I think somehow showing the degree of utilization for isolates give the reader a complete view of the data. Providing some type of range for delineating between +,-, and +/- would be sufficient for addressing the issue.

• The authors state, “Bacillus amyloliquefaciens and B. subtilis have been recognized as the most dominant bacteria in bees foraging for food from rape flowers” and cite Wang et al., 2015. (see lines 318 – 319). The results the authors are referring to were actually presented in a previous publication by Wang et al. that characterized the diversity of bacteria in the honey stomach of bees during the rape blooming season, where Bacillus species were observed to dominate the dataset1. The publication the authors have cited is follow-up paper that mentions said results, but does not provided any such data. Please make certain to cite primary sources accordingly.
1. Wang, M., Xu, H., Yin, G., Zhao, W., & He, S. (2014). Diversity of bacteria in the honey stomach of Apis cerana and Apis mellifera during the rape blooming period. Chin. J. Appl. Entomol, 51, 1567-1575.

• For citations pertaining to lines 313 – 317, please clarify if the said strains of Bacillus were isolated from the honeybee gut and honey or the honeybee and/or honey. Currently the way the sentences is structured it is saying all of the strains referenced are found in both. If this is true then disregard the comment.

• This section of the manuscript needs a bit of work. Similar to the introduction, all the structural elements are there but there is an absence of cohesiveness. The main reason for this is likely due to the absences of a defined goal/hypothesis for the study. If the authors add a brief paragraph in the beginning of the discussion recapping what was known, what was unknown, and the new information acquired by this study, it would setup the necessary framework for the paragraphs that follow.

Experimental design

• Could you clarify the purpose of the TPC analysis for each of the hive components? Currently, I am under the impression that the goal of the TPC analysis was to demonstrate the bacterial load (total bacterial community) for each hive components. Is this true? If so, this analysis is likely not the best way to address this question. Is it possible that the TPC analysis would have been more inclusive of other microbes if additional media types and growth conditions were included? I ask because based on the results presented, it seems like the conditions provided were highly selective for bacteria in the genus Bacillus as nearly all of the isolates cultured were classified as such (37/41). Would you observe the same results with a different media type? Furthermore, when you say the hive components had “low TPC levels” what exactly do you mean? And what are you comparing these values to? Please clarify.

• For each isolate, how many biological replicates were conducted for Biolog GenIIIPlate assay?

• For the antimicrobial activity assay – is it possible some of strains show no or little antimicrobial activity because physical interaction with another microbe is required before antimicrobial production is activated? If the authors could address this in the manuscript it would be greatly appreciated. Furthermore, is there a reason endogenous microbes of the H. itama microbiome weren’t examined?

Validity of the findings

• Where are the statistical analyses of datasets? For example, claims made regarding TS1 (lines 219 – 226) should have stats to backup interpretations of the reported results.

• I would urge the authors to be cautious with their interpretations of the data collected in this study. The authors claim that in vitro metabolic profiling of Bacillus isolates suggest the presence of microbes in bee products indicating the availability of nutrients to support microbial viability (Lines 300 – 304). First, all of the data collected in this study were under in vitro conditions, which provide a glimpse into the metabolic potential of microbes, but can not be directly generalized to the function of the microbe during in vivo conditions. Furthermore, the data collected doesn’t speak to the nutrient availability in hive components or the microbial viability within said environments. Currently these ideas are not addressed in the discussion. The authors should add text to provide adequate context to their study.

Additional comments

The manuscript by Ngalimat et al characterizes Bacillus species isolated from hive components (honey, bee bread, and propolis) from the hive of the stingless bee, Heterotrigona itama. Currently, much is known about the microbiome of Apis species, especially Apis mellifera (the honey bee), with particular focus on microbiome composition, function, and the various factors that influence these processes. Overall, studies on the microbiome of other Apis species are somewhat limited compared to A. mellifera. The study by Ngalimat et al focuses on the isolation and characterization of bacterial species within the genus Bacillus that are often found associated with H. itama. The authors’ objective for this study was to elucidate the potential functional role of Bacillus species in hive components. Through utilizing culturing efforts the authors isolated bacteria, many of which were Bacillus species, from honey, bee bread, and propolis. The authors go on to characterize, in vitro, the metabolic and antimicrobial activity of a subset of isolates (N = 11). Initially I am intrigued by this study and the efforts attempted by Ngalimat et al. Overall, there are many positive attributes to the manuscript. The overall composition of manuscript is in great shape. The text is concise and easy to follow and read. The research design is sound and straightforward, and the results presented are convincing. Concerns regarding the manuscript primarily arise due to: 1) Inadequate development of the rational driving the research objective(s) (see comments on the introduction and discussion sections), 2) Absence of statistical analyses, 3) Issues with the arrangement of figures, 4) Issues with citations.

Reviewer 2 ·

Basic reporting

In general the paper has been written relatively well, with the basic idea behind of the work that was carried adequately highlighted. Though, the authors need to be mindful the grammatical errors in the paper, which I believe can be further improved after a few additional rounds of checkings. Certain sentences require attention, for instance:

1. “:However, their biological functions are poorly understood and mostly indicated it as a microbial contaminant originated from pollen, flower, bees’ digestive tract, dirt, dust and …”
2. “In addition, Bacillus species that 82 isolated from honeybee honey produced antimicrobial compounds (Zhao…”

Experimental design

In general, the work has been well-described.

Please check the sentences:

Amend the sentence “Dilution of pure cultures were prepared by performing a ten fold serial dilution in a range of 10-1 on each sample followed by incubation at 37°C with shaking at 150 rpm for 24 h.”

The sentence in the section for the “Determination of antimicrobial activity” should be amended to “The 6 mm diameter wells were cut on the agar plate using ??? specify what used to cut out the wells… and filled with 100 μl of supernatant of each isolates obtained by centrifugation at 6000 × g for 15 min at 4°C.”

This sections requires doublecking for grammatical errors

Validity of the findings

The findings appears to agree with many of reported literature, with a few novel findings that would be useful to improve the body of knowledge with regards to antimicrobial potential of the H. itama stingless bees, as well as the bee products.

I believe the findings in this study is very interesting.

However, please doublecheck the following sentences for improvement of grammar etc:

“The TPC ofpropolis ranges from 6.3 × 103 cfu/g (Ladang 10 UPM) to 1.8 × 104 cfu/g (Yayasan Al223 Jenderami) while those recorded for honey samples ranges from 0.0 cfu/g (Ladang nangka PASFA) to 8.0 × 103 cfu/g 224 (Yayasan Al-Jenderami).”.

Break this sentence, into two separate sentences for clarity. The sentence should be improved as the following: “According to the results, 41 isolates were recovered from H. itama nest products. A total of 37 screened isolates were identified as bacteria from the genus Bacillus, as analysed by Gram and endospore staining as well as catalase production (Table S2).”

Amend the grammar. It should be “The results indicated that each isolate has a different profile of substrates utilisation and chemical sensitivity as determined by Biolog…”

Additional comments

The findings in this paper, in my opinion, merits publication in PeerJ. However, it believe the paper is publication ready following minor corrections.

Reviewer 3 ·

Basic reporting

I suggest the authors get editing help from someone with full professional proficiency in English. Currently, some of the sentences are difficult to understand, for example: ; 81-82; 353-355;

Experimental design

The Bacillus cereus comprises a highly metabolically diverse group of bacteria (Cepeus et al., 2013). If the objective of this study was the prospecting for strains that have desired properties, all the 37 strains should have been evaluated.

Validity of the findings

no comment

Additional comments

The manuscript "Isolation and characterisation of Bacillus spp. from stingless bee, Heterotrigona itama honey, bee bread and propolis" describes the phenotype of 37 Bacillus strains isolated from stingless bees' products.
The methods used in this study are appropriate and the data and statistical analysis are sound.
My major concerns are:

1. The current English grammar and style is not appropriate.

2. There is no clarity as to the main objective of this study.

3. In the introduction the authors wrote: "Regardless of plenty antimicrobial activities, some microbes such as Bacillus species still can survive in bee products (Lee et al., 2015) due to the robustness of bacterial endospore which is resistant to heat, radiation and chemicals (Nicholson, 2002; Setlow, 2014)." Based on this, I have some questions:
- Are Bacillus cells active in the bee products or only found in a dormant state?
- Why looking for Bacillus spp. in bee products? Why not other habitats, such as pollen, in which that they could be found in higher abundance and diversity?

Minor comments:

- Were the enzyme activity assays performed in replicates? If yes, please add this information to the Methods section.

---

## Round 0.2 · Major Revisions

The reviewers have spent a considerable amount of time and attention helping you with this manuscript. Please do follow their recommendations and respond to each reviewer critique substantively.

Reviewer 1 ·

Basic reporting

Introduction:

In general, the major issue with the introduction is the lack of a clear hypothesis. It seems as though the authors are attempting to develop a manuscript around the following ideas: 1) bee products have beneficial properties, 2) some of the bacteria found in bee products are beneficial against pathogens, 3) Other bacteria besides LABs, might contribute beneficial properties and need to be investigated. I would urge the authors to edit the introduction so that it is simple, concise, and develops the points I have described above.

Lines 51 – 54: This sentence is extremely confusing. Rephrase for clarity.

Line 67 – 68: What do you mean by primary and secondary contamination?

Lines 65 – 66: What “products” are you referencing to? The Lee et al. 2015 paper is a study on the honey bee gut microbiome. There are other studies in the field that have cultured microbes from honey, pollen, and beebread.

Line 80 – 83: There are many studies that have begun to address this hypothesis, especially for the processing of pollen to beebread. I would suggest the authors seek out those studies and cite them as they are pertinent to this paper.

Figures & Tables:

Figure 1: please add text in the figure legend that indicates that bolded text indicates bacteria isolated in this study.

Table 2: Please put the data for B. oleronius PD3 and B. Wealsonii PD4 in the table. I understand that negative results were reported for these strains, but negative results are still results and should be reported regardless.

Table 3: Report data for isolates with negative results (see comment about Table 2).

Table S1: What are the extremely small numbers next to the values? Exponents? Either way, the text is far too small to read. The figure legend indicates statistics were performed on the data. What kind if statistics were performed? T-test? ANOVA? Was the data normally distributed? This information is not provided in the figure legend or the methods section.

File S1 and S2: In these files there are “A” and “B” columns for each isolate. What does that mean? Are these biological replicates? There is no indication of the purpose of this nomenclature. Please clarify.

Experimental design

No Comment

Validity of the findings

Results:

Lines 240 – 242: I’m a bit confused by this sentence, “Among the three nest products, propolis showed the highest TPC value……while the lowest was honey (0.0 cfu/g).” (Lines 240 – 242). Why is honey the lowest? Beebread also had a TPC value of 0.0 cfu/g for one location. Why isn’t beebread mentioned?

Discussion:

The organization of the discussion needs a bit a work. In its current state, each paragraph seems like a separate entity without any connectivity. This makes the discussion challenging to read. I would encourage the authors to revise this section into a more cohesive text. Finally, it is totally acceptable for the authors to hypothesize the context of their finds, but the authors should also note some of the limitations to the study and their ability to speculate in vivo functionality.

Lines 324 – 328: These introductory sentences are not very helpful for this paragraph that focuses on the TPC data.

Lines 324 – 326: The authors state that “interactions of bee and bacteria are still unclear”, that is not exactly true. There is a large body of literature that addresses just that topic in the honey bee. Perhaps the authors are meaning to reference the stingless bee?

Lines 326 – 328: “…, we partially discovered these enormous characteristics.” – I’m not sure if “enormous” is the right wording. I would revise this sentence.

Lines 345 – 354: In this paragraph the authors discuss some of the similar species of Bacilli observed in stingless bees and the honey bees. Additional references regarding the honey bee microbiome need to be cited. For example, a recent study by Tarpy, Mattilla, and Newton, 2015 characterize the gut microbiome of the honey bee queen. This study should be cited when the authors discuss the queen microbiome (Lines 348 – 352). There is a plethora of other studies that have characterize the microbial communities associated with the honey bee. I would strongly recommend the authors thoroughly examine the literature.

Tarpy, David R., Heather R. Mattila, and Irene LG Newton. "Development of the honey bee gut microbiome throughout the queen-rearing process." Appl. Environ. Microbiol. 81.9 (2015): 3182-3191.

Lines 369 – 370: The authors speculate that based on the phenotypic substrate utilization assay, the patterns of utilization suggest that the bacteria analyzed might be involved in the formation of bee products. Based purely on in vitro data, I would urge the authors to be cautious with this statement. Currently, it is unclear if the bacteria isolated are active when present in bee products. As suggested by the authors (Lines 69 – 71) these bacteria might be dormant in vivo. If so, the microbes functional potential might be extremely reduced. I would encourage the authors to discuss some of the limitation to their approach. Additionally, it could also be hypothesized that the bacteria could be involved in the degradation of bee products instead of the “formation” of bee products.

Lines 328 – 329: The authors state “In this study, low level of TPC and limited types of bacteria were found.”. It should be noted that the limited profile of bacteria isolated could be due to cultivation on a single medium type and growth conditions.

Additional comments

Overall the authors have made substantial edits that have significantly improved upon the quality of the manuscript. The authors have addressed many of the concerns brought to their attention in the previous review process. For the most part, the methods and results sections are in great shape, as are the figures and tables (a few minor adjustments are needed). Nevertheless, issues remain in the current version of the manuscript. The authors need to seek out additional assistance composing the introduction and discussion sections. Currently, these sections are the most difficult to read and understand (see specific comments in the “basic reporting” section).

Reviewer 3 ·

Basic reporting

no comment.

Experimental design

no comment.

Validity of the findings

no comment.

Additional comments

Minor comments:

Line 29: change "from phylum Fermicutes (...)" to "from the phyla Firmicutes (...)". Firmicutes needs to be corrected in the entire manuscript. Including figures;
Line 48: change to "Stingless bees are important pollinating agents (...)";
Line 62: change "Stingless bee products such as honey are" to "Stingless bee products, such as honey, are". "such as XXX" has to be between commas. This needs to be corrected in the entire manuscript;
Lin 96: change "Investigations on the association of bacteria in stingless bee products have been focused mostly (...)" to "Investigations of bacteria associated to stingless bee products have been focused mostly (...)";
Line 399: “limited and common bacterial species” is confusing.

---

## Round 0.3 · accepted · Accept

You have substantively responded to the reviewer concerns and the manuscript is much improved.